# A Nationwide Study of the “July Effect” Concerning Postpartum Hemorrhage and Its Risk Factors at Teaching Hospitals across the United States

**DOI:** 10.3390/healthcare11060788

**Published:** 2023-03-07

**Authors:** Zahra Shahin, Gulzar H. Shah, Bettye A. Apenteng, Kristie Waterfield, Hani Samawi

**Affiliations:** 1Department of Health Policy and Community Health, Jiann-Ping Hsu College of Public Health, Georgia Southern University, P.O. Box 8015, Statesboro, GA 30458, USA; 2Department of Biostatistics, Epidemiology and Environmental Health Sciences, Jiann-Ping Hsu College of Public Health, Georgia Southern University, P.O. Box 8015, Statesboro, GA 30458, USA

**Keywords:** postpartum hemorrhage, causes of postpartum hemorrhage, management, July effect, patient safety

## Abstract

Objective To assess the “July effect” and the risk of postpartum hemorrhage (PPH) and its risk factors across the U.S. teaching hospitals. Method This study used the 2018 Nationwide Inpatient Sample (NIS) and included 2,056,359 of 2,879,924 single live-birth hospitalizations with low-risk pregnancies across the U.S. teaching hospitals. The International Classification of Diseases, Tenth Revision (ICD-10) from the American Academy of Professional Coders (AAPC) medical coding was used to identify PPH and other study variables. Multivariable logistic regression models were used to compare the adjusted odds of PPH risk in the first and second quarters of the academic year vs. the second half of the academic year. Results Postpartum hemorrhage occurred in approximately 4.19% of the sample. We observed an increase in the adjusted odds of PPH during July through September (adjusted odds ratios (AOR), 1.05; confidence interval (CI), 1.02–1.10) and October through December (AOR, 1.07; CI, 1.04–1.12) compared to the second half of the academic year (January to June). Conclusions This study showed a significant “July effect” concerning PPH. However, given the mixed results concerning maternal outcomes at the time of childbirth other than PPH, more research is needed to investigate the “July effect” on the outcomes of the third stage of labor. This study’s findings have important implications for patient safety interventions concerning MCH.

## 1. Introduction

Postpartum hemorrhage (PPH) is an obstetric emergency and one of the contributing factors to maternal morbidity and mortality [1,2]. Researchers categorized four “T’s”, including Tune (uterine atony), Tissue (retained tissue, invasive placenta), Trauma (uterine rupture, genital tract laceration, and uterine inversion), and Thrombin (coagulation abnormalities, disseminated intravascular coagulation) as the causes of PPH [3]. A population-based cohort study observed that prolonged labor, episiotomy, delayed initial care for PPH, specifically administration of oxytocin more than 10–20 min after PPH diagnosis, and waiting more than 10 min to call for additional assistance (an obstetrician and an anesthesiologist), increased the risk of severe PPH [4]. Studies also found that PPH increases with the third stage duration of 20 min, and the risk of severe PPH rises with the third stage duration of 23–25 min [5,6]. Every ten minutes of extra delay has been associated with an increased adjusted odds of PPH risk of 1.11 (1.02–1.21) and 1.14 (1.03–1.27) for the risk of severe PPH [7].

PPH has also been associated with poor obstetric practice and clinical management of postpartum hemorrhage. Delayed diagnosis of severe hemorrhage, poor team communication, limited access to timely and quality care, delayed transfusion, and inadequate access to resources such as blood products have been shown to be contributing factors that can increase the risk of severe PPH [8,9,10].

Active management of the third stage of labor is critical in reducing PPH occurrence but is influenced by many factors, including standard guidelines and protocols, quick service accessibility, availability of resources (equipment and staff), and quality of care services [9,10,11,12].

Researchers suggested that the arrival of new and inexperienced residents and interns may increase adverse patient outcomes [13,14,15]. July is the month that the academic year begins, experienced residents and fellows depart, and new residents and interns arrive [13]. Some studies observed increased adverse patient outcomes or inefficient care during this transition, known as the “July effect” [13,14,15]. A previous study using 2018 Nationwide Inpatient Sample (NIS) data to examine the association of the risk of PPH with hospital characteristics found that teaching hospitals across the U.S. had the highest adjusted odds of increased risks of PPH [16]. Therefore, the same data (the NIS 2018 data) was used to assess the “July effect” on the increased risks of PPH across U.S. teaching hospitals. In addition, the study investigated the “July effect” concerning factors that increase the risk of PPH, such as failed induction of labor, puerperal infections, other obstetric trauma, and perineal laceration. This study is the first to focus on the correlation between the “July effect” and PPH risk across U.S. teaching hospitals.

## 2. Methods

This retrospective cohort study used the 2018 National Inpatient Sample (NIS) from the Healthcare Cost and Utilization Project’s (HCUP) databases. NIS is a hospital inpatient stay database derived from hospitals’ billing data from statewide data organizations across the U.S. The NIS database includes all patient discharges from community hospitals in the U.S.; however, since 2012, it has not included rehabilitation hospitals or long-term acute care hospitals. The NIS data provides the opportunity to create national and regional estimates by analyzing weighted data to produce accurate, unbiased results and demonstrate larger universe data [17]. It includes clinical and resource use information from discharge abstracts and contains data for all hospital stays, regardless of payer. The NIS uses the ICD-10-CM/PCS coding system to report a full calendar year of data with diagnosis and procedure codes. At the beginning of the 2016 data year, the NIS coding schema changed from the ICD-9-CM diagnosis codes to the ICD-10-CM/PCS codes [18].

### 2.1. Study Population

A sample of women aged ≤19–54 in the third stage of labor with an index for postpartum hemorrhage (PPH) was selected using the NIS data from 1 January 2018 to 30 December 2018 across the U.S. teaching hospitals. This study included women with low-risk clinical conditions associated with maternal hemorrhage and excluded cases with previous c-sections, women with intrapartum hemorrhage, placental abruption, placenta previa, and high-risk cases in obstetrics because of their high-risk clinical conditions associated with maternal hemorrhage.

### 2.2. Measures or Variables

The International Classification of Diseases, Tenth Revision (ICD-10) from the American Academy of Professional Coders (AAPC) for medical coding was used and identified codes O61, O63, O70, O71, O72, and O86 for failed induction of labor, prolonged labor, perineal laceration during delivery, other obstetric trauma, postpartum hemorrhage (PPH), and puerperal infections, respectively. Other codes, including O0993, O432, O43213, O43223, O43233, O468X3, O458X3, O4693, O610, O611, O618, O619, O63, O630, O702, O703, O710, O719, O8611, O8612, O8619, Z3800, and Z3801, were selected for further analysis (Appendix A) [18,19].

The main dependent variable of interest for this study was PPH, and the corresponding ICD-10 code was identified as O72. The independent variable included the “July effect”, which was operationalized by categorizing the variable month (month of hospitalization) into three groups: (1) January to June (the second half of the academic year); (2) July to September (the first quarter of the academic year); and (3) October to December (the second quarter of the academic year).

Covariates included the assessed region of the teaching hospital location, maternal age, race, or ethnicity, and variables that increase the risks of PPH, including failed induction of labor, prolonged labor, puerperal infections, other obstetric trauma, and perineal laceration during delivery. Maternal age was categorized as ≤19, 20–34, and 35–54 years of age. Maternal race and ethnicity were defined based on HCUP coding as follows: (1) White, (2) Black, (3) Hispanic, (4) Asian or Pacific Islander, (5) Native American, and (6) Other [18].

### 2.3. Statistical Analysis

We performed all analyses with STATA software version 16.1 [19]. Five multivariable logistic regression models were used to estimate the adjusted odds ratio and 95% confidence intervals (CI) of PPH and factors related to increasing PPH risk while controlling for regions of teaching hospitals, race, and age. A sixth multivariable logistic regression was performed to estimate the adjusted odds ratio and 95% confidence intervals (CI) of PPH in the first and second quarters with the second half of the academic year while controlling for regions of teaching hospitals, age, race, or ethnicity, plus PPH risk factors (including failed induction of labor, other obstetric trauma, perineal lacerations (3rd and 4th degree), and puerperal infections) [8,20].

This study was exempted by Georgia Southern University’s institutional review board (IRB) from full board review as it uses de-identified secondary data.

## 3. Results

In 2018, there were 2,879,924 single live-birth hospitalizations with low-risk clinical conditions associated with maternal hemorrhage among women ages ≤19–54 across the U.S. teaching hospitals. Of these, 48.74% were performed in the months of January through June, 26.27% from July through September, and 24.99% from October through December. Approximately 4.2% of the women had postpartum hemorrhage (PPH). In the sample population, women’s races included White 49.80%, Black 15.84%, Hispanic 21.81%, Asian or Pacific Islander 6.96%, Native-American 0.54%, and other 5.05%. The age group of ≤19 accounted for approximately 5.19%, the age group of 20–34 accounted for almost 94.60%, and the age group of 35–54 accounted for only 0.21%.

### 3.1. Test of July Effect: Multivariable Logistic Regression of PPH and Its Risk Factors

We performed five multivariable logistic regressions to test the association of PPH risk and each of its risk factors separately with the month of delivery (to indicate the “July effect”) while holding regions of teaching hospitals, race, and age constant (see Table 1). Results showed that compared to women who delivered during January through June, those who delivered during July through September had significantly higher adjusted odds of PPH (AOR, 1.05; CI, 1.02–1.10). Compared to deliveries from January through June, the adjusted odds for PPH were also significantly higher during the months of October through December (AOR, 1.07; CI, 1.04–1.12). However, there were no observed discharge month effects for the assessed PPH risk factors—failed induction of labor, puerperal infections, perineal laceration, and obstetric trauma.

Results for regions of teaching hospital locations showed that compared to Northeastern teaching hospitals, women who delivered at Midwest teaching hospitals had lower adjusted odds of puerperal infection (AOR = 0.62; CI = 0.42–0.78) but higher adjusted odds of PPH (AOR = 1.21; CI = 1.01–1.28) and other obstetric trauma (AOR = 1.26; CI = 1.08–1.46). Women who delivered at teaching hospitals located in the South region had lower adjusted odds of PPH (AOR = 0.78; CI = 0.71-.89), failed induction of labor (AOR = 0.55; CI = 0.45–0.74), and puerperal infections (AOR = 0.49; CI = 0.41–0.61) compared to those who delivered at Northeastern hospitals. Women who delivered at West teaching hospitals had higher adjusted odds of PPH (AOR = 1.20; CI = 1.07–1.37) and other obstetric trauma (AOR = 1.24; CI = 1.06–1.44) but lower adjusted odds of puerperal infections (AOR = 0.67; CI = 0.53–0.83) compared to Northeastern teaching hospitals.

Results for race and age as control variables found that all races compared to White women had significantly higher adjusted odds of PPH and puerperal infections, and the adjusted odds were the highest for Native Americans [(AOR = 1.61; CI = 1.32–1.98) and (AOR = 2.4; CI = 1.54–3.87), respectively]. However, different results were observed for race and other factors that increase the risk of PPH. For instance, results showed that compared to being White, being Black and Hispanic decreased the adjusted odds of failed induction of labor [(AOR = 0.60; CI = 0.52–0.68) and (AOR = AOR = 0.72; CI = 0.64–0.81), respectively)] and perineal laceration [(AOR = 0.52; CI = 0.47–0.56 and (AOR = 0.64; CI = 0.59–0.69), respectively)] and, being Asian or Pacific Islander, increased adjusted odds of failed induction of labor, puerperal infections, and perineal laceration [(AOR = 1.28; CI = 1.08–1.52), (AOR = 2.2; CI = 1.71–2.81), and (AOR = 2.23; CI = 2.03–2.45), respectively].

Results also showed that each one-year increase in age decreased the adjusted odds of PPH risk, puerperal infections, perineal laceration, and other obstetric trauma by 13%, 53%, 11%, and 49%, respectively (see Table 1).

### 3.2. Multivariable Logistic Regression for PPH: Adjusting for PPH Risk Factors

A multivariable logistic regression was performed to estimate adjusted odds ratios for PPH risk with the month of delivery (to indicate the “July effect”) while holding race, age, failed induction of labor, other obstetric trauma, perineal laceration during delivery, and puerperal infection constant (Table 2). Compared to women who delivered during the second half of the academic year (January–June), those who delivered during the first quarter of the academic year (July–September) and the second academic year (October–December) had higher adjusted odds of PPH, and the adjusted odds were the highest for delivery during the second quarter of the academic year (October–December): 1.07 (1.03–1.12) (*p* < 0.0001) than the first quarter of the academic year: 1.05 (1.02–1.10) (*p* = 0.004).

Compared to Northeastern teaching hospitals, women who delivered in South teaching hospitals had lower adjusted odds of postpartum hemorrhage (AOR = 0.79; CI = 0.69–0.91), while women who delivered in Midwest and West teaching hospitals had higher adjusted odds of PPH [(AOR = 1.21; CI = 1.06–1.38) and (AOR = 1.20; CI = 1.05–1.35), respectively.

Results indicated that, compared to White women, all other race groups had increased adjusted odds of PPH, but the adjusted odds were the highest for Native Americans compared to other races (AOR = 1.62; CI = 1.32–1.97). Each one-year increase in age decreased the adjusted odds of PPH (AOR = 0.91; CI = 0.84–0.98).

Compared to its absence, the presence of PPH risk factors, including perineal laceration, other obstetric trauma, puerperal infections, and failed induction of labor, was associated with increased adjusted odds of PPH of 1.98 (CI = 1.78–2.18), 2.61 (CI = 2.46–2.83), 5.44 (CI = 4.75–6.23), and 1.97 (CI = 1.76–2.23), respectively. The odds were the highest for puerperal infections (AOR = 5.44).

## 4. Discussion

This study used the 2018 National Inpatient Sample (NIS) from the HCUP database to compare the risk of PPH and its risk factors in the first and second quarters with the second half of the academic year to assess the effect of the “July effect” in teaching hospitals across the U.S. The analysis showed increased adjusted odds of PPH in the first and second quarters of the academic year compared with the second half of the academic year. The adjusted odds were the highest for the second quarter of the academic year (October–December) (*p* < 0.0001) than the first quarter of the academic year (July–September) (*p* < 0.004). There is no explanation for these differences. However, researchers suggested that one probability could be that in the second quarter of the academic year, trainees have more autonomy and less supervision in their decision-making regarding the status of patient care compared to the first quarter of the academic year and that sometimes can also lead to an increase in adverse patient health outcomes [21]. This study found no statistically significant differences for factors that increase the risk of PPH in the first and second quarters vs. the second half of the academic year. However, increased adjusted odds of other obstetric trauma in Midwest and West teaching hospitals and puerperal infections in Northeastern teaching hospitals need further investigation.

Although this project provided support for an association between the “July effect” and the increased risk of PPH, most literature on obstetric and gynecologic surgery reviewed by this study found no significant associations between the “July effect” and the third stage of labor outcomes [22,23,24,25,26], and few studies found an increase in length of hospital stay (LOS) [20] and cesarean rates [26], and only one study found minimum support for the “July effect” on delivery complications, and that was before July 2003 [27].

It should be noted that the method used by this study to assess the “July effect” on the increased adjusted odds of PPH was different from methods used in the cited literature. For instance, studies either compared the obstetric outcomes in the first quarter with the second quarter [24] or for each month of the academic year separately [25] or the first quarter with the last quarter [26]. One study that used NIS data also compared the month of July with the months of August to June [22]. The only study that compared early (July or August) versus late academic year (April or May) found minimum support for the “July effect”, and that was a retrospective study of 628,414 singleton births in Washington State from 1987 to 2012. The Washington State study estimated the risk of maternal peripartum complications and found a small “July effect” on maternal peripartum complications before July 2003 and before the establishment of work hour reform (residents work 80 h per week), but no significant differences were observed between teaching and non-teaching hospitals after July 2003 and work hour reform [27,28]. Our study finds the “July effect” may extend to the first half of the new academic year.

### 4.1. Limitation

The study’s limitations included its inability to establish a causal effect due to its observational, cross-sectional design. Due to the limitations of the available data, information was unavailable on whether residents, interns, or other birth attendants were specifically involved in each discharge. Contextual information about residents and the residency programs was also not assessed, including the status of the changeover system and supervision of training skills and experience. Research should examine longitudinal data to provide more evidence for the July effect.

### 4.2. Recommendations

Despite medical advances, PPH remains one of the challenges of obstetric care services. Therefore, management of PPH regarding patient safety needs to be carried out throughout the pregnancy and with careful and intentional management during the third stage of labor. Risk factors for PPH should be assessed during prenatal care and the high-risk case process. All the initial assessment and patient resuscitation has to be conducted within the first 15 min of a PPH occurrence. A multidisciplinary team must be vigilant in the event of a PPH occurrence [29]. Essential components of the management of the third stage of labor include the availability of required medications, blood production, and standard protocols and instructions in delivery units regarding hemorrhage [10]. Protocols and guidelines should provide the knowledge and tools necessary for the obstetric team’s early detection and rapid evaluation of PPH. However, successful implementation of protocols and guidelines requires active management, multidisciplinary team cooperation, available resources, constant surveillance, and staff training programs regarding protocol performance [29].

While this study does not establish a causal effect of the “July effect” on patient outcomes, the findings show an increased risk of PPH in the first half of the academic year. These findings have implications for improving the teaching and care environments in teaching hospitals. Though it should be noted that not all trainees at the same level demonstrate the same knowledge and skills, medical schools must warrant that new interns and residents are trained and prepared with new protocols and practices before taking on new roles in the clinical setting, and that should be continued through the changeover [16]. The findings suggest careful assessment of collected data to evaluate and investigate the third stage of labor outcomes regarding PPH in the early and late academic year to institute more rigorous roles and protocols regarding the supervision of new trainees and prevent adverse obstetric care outcomes [21]. Policies should focus on the future of new trainees’ education and practice reform in ways to identify weaknesses and challenges in practice and improve patient safety at teaching hospitals [27].

## 5. Conclusions

Overall, the root causes of PPH conditions should be investigated to determine deficiencies in the system and identify whether they are related to maternal risk factors or to PPH management by considering the availability of resources, including skilled birth attendants, multi-professional team collaboration, and medical services such as blood products and medications, and how they can be improved [10]. This study found an increased risk of PPH in the first half of the academic year in teaching hospitals, suggesting the need for thorough supervision of the implementation of hospital policies and protocols regarding maternal safety and training programs in teaching hospitals. Previous studies have found that residency work-hour reform reduced medical errors and fatigue and improved residents’ quality of life, productivity, and patient health outcomes [27]. Therefore, it is critical to scrutinize whether a higher incidence of PPH is related to residents’ work hours, defective procedural techniques, or overall preprocedural care and unfamiliarity with hospital procedures and practices [30]. It is important to note that improving monitoring and practical clinical training are essential to mitigate the impacts of changeover [16]. In addition, more research is needed to focus on medication errors, delayed or incorrect diagnoses, and the vulnerabilities of patients to adverse outcomes.

## Figures and Tables

**Table 1 healthcare-11-00788-t001:** Logistic regression of Postpartum Hemorrhage and Factors Increased PPH Risk across the U.S. Regions and Hospital Characteristics.

Variables	Postpartum Hemorrhage		Failed Induction of Labor		Puerperal Infections		Perineal Laceration		Other Obstetric Trauma	
	AOR and CI	*p*-Value	AOR and CI	*p*-Value	AOR and CI	*p*-Value	AOR and CI	*p*-Value	AOR and CI	*p*-Value
Month Reference (Jan–June)										
July–September	1.05 (1.02–1.10)	**0.004**	0.96 (0.88–1.04)	0.313	1.03 (0.92–1.17)	0.546	0.96 (0.91–1.02)	0.181	1.02 (0.97–1.08)	0.306
October–December	1.07 (1.04–1.12)	**<0.001**	0.98 (0.89–1.07)	0.660	1.08 (0.96–1.23)	0.165	1.01 (0.96–1.06)	0.743	1.04 (0.98–1.01)	0.114
Region Reference (Northeastern)										
Midwest	1.21 (1.01–1.28)	**0.025**	0.92 (0.72–1.18)	0.537	0.62 (0.49–0.78)	**<0.001**	1.09 (0.98–1.21)	0.098	1.26 (1.08–1.46)	**0.002**
South	0.78 (0.71–.89)	**<0.001**	0.55 (0.45–0.74)	**<0.001**	0.49 (0.41–0.61)	**<0.001**	1.02 (0.93–1.12)	0.644	1.07 (0.93–1.22)	0.309
West	1.20 (1.07–1.37)	**0.002**	1.29 (0.99–0.1.67)	0.051	0.67 (0.53–0.83)	**<0.001**	1.04 (0.93–1.16)	0.448	1.24 (1.06–1.44)	**0.005**
RACE Reference (White)										
Black	1.21 (1.13–1.29)	**<0.001**	0.60 (0.52–0.68)	**<0.001**	2.51 (2.16–2.91)	**<0.001**	0.52 (0.47–0.56)	**<0.001**	0.95 (0.88–1.03)	0.284
Hispanic	1.34 (1.22–1.46)	**<0.001**	0.72 (0.64–0.81)	**<0.001**	2.09 (1.80–2.44)	**<0.001**	0.64 (0.59–0.69)	**<0.001**	1.07 (0.99–1.16)	0.052
Asian or Pacific Islander	1.44 (1.31–1.57)	**<0.001**	1.28 (1.08–1.52)	**0.004**	2.19 (1.71–2.81)	**<0.001**	2.23 (2.03–2.45)	**<0.001**	0.91 (0.82–1.01)	0.068
Native American	1.61 (1.32–1.98)	**<0.001**	0.76 (0.40–1.22)	0.214	2.44 (1.54–3.87)	**<0.001**	0.80 (0.51–1.25)	0.330	1.21 (0.92–1.58)	0.155
Other	1.17 (1.06–1.29)	**0.001**	0.77 (0.63–0.93)	**0.014**	2.32 (1.87–2.89)	**<0.001**	1.15 (1.03–1.28)	**<0.001**	0.99 (089–1.11)	0.955
AGE	0.87 (0.81–0.93)	**<0.001**	1.06 (0.90–1.25)	0.468	0.47 (0.40–.55)	**<0.001**	0.89 (0.80–0.98)	**<0.001**	0.51 (0.48–0.56)	**<0.001**

Note. *p*-value set at *p* < 0.01 statistically significant and *p* < 0.001 highly statistically significant.

**Table 2 healthcare-11-00788-t002:** Logistic Regression of Postpartum Hemorrhage: Adjusting for PPH Risk Factors.

Variables	Postpartum Hemorrhage	
	AOR and CI	*p*-Value
Month Reference (Jan–June)		
July–September	1.06 (1.01–1.09)	**0.005**
October–December	1.07 (1.03–1.12)	**<0.001**
Region Reference (Northeastern)		
Midwest	1.21 (1.06–1.38)	**0.004**
South	0.79 (0.69-.91)	**0.001**
West	1.20 (1.05–1.37)	**0.006**
RACE Reference (White)		
Black	1.22 (1.14–1.30)	**<0.001**
Hispanic	1.34 (1.22–1.45)	**<0.001**
Asian or Pacific Islander	1.39 (1.27–1.52)	**<0.001**
Native American	1.61 (1.32-.197)	**<0.001**
Other	1.16 (1.05–1.27)	**0.001**
AGE	0.91 (0.85–0.98)	**0.013**
Perineal laceration	1.98 (1.78–2.18)	**<0.001**
Other obstetric trauma	2.61 (2.46–2.83)	**<0.001**
Puerperal infections	5.44 (4.75–6.23)	**<0.001**
Failed induction of labor	1.97 (1.76–2.23)	**<0.001**

Note. *p*-value set at *p* < 0.01 statistically significant and *p* < 0.001 highly statistically significant.

## Data Availability

Not applicable.

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
