# Peer review of "A Nationwide Study of the “July Effect” Concerning Postpartum Hemorrhage and Its Risk Factors at Teaching Hospitals across the United States"

_healthcare, 2023, doi:10.3390/healthcare11060788_

Round 1

Reviewer 1 Report

In the last paragraph of the introduction should go the objectives.

An important variable that has not been assessed in the study is who attended the delivery, whether it was a midwife or gynecologist or, on the contrary, a resident or intern and if so, from what year. If this information is available, it should appear in the study, if not, it should be mentioned in the limitations section.

Author Response

Reviewer 1

Reviewer Comment/Suggestion

Page and line number of comment to be addressed

Explanation of how it was addressed by the authors

In the last paragraph of the introduction should go the objectives.

Page 4, introduction, Last paragraph

Thank you for your suggestions and comments.

We revised this paragraph to address your comment.

An important variable that has not been assessed in the study is who attended the delivery, whether it was a midwife or gynecologist or, on the contrary, a resident or intern and if so, from what year, and if this information is available, it should appear in the study, if not, it should be mentioned in the limitations section

Page 13, limitations:  line 3-4

In the NIS data, there is no information regarding birth attendants. However, we edited this section and added other birth attendants to the sentence to address your comment.

Thank you.

Reviewer 2 Report

  1. Thank you for submitting the manuscript.
  2. line 74-

  3. Study population :low-risk sample of women aged ≤19- 54 . Kindly clarify , do you think pregnancy below age of 19 or above 40 can be labelled as low risk.

  4. Table -1: Age as variable  should also have a reference for comparison
  5. Recommendations and conclusions have to be well defined . References quoted in these sections are not appropriate.

Author Response

Reviewer 2

Reviewer Comment/Suggestion

Page and line number of comment to be addressed

Explanation of how it was addressed by the authors

line 74-

Study population :low-risk sample of women aged ≤19- 54 . Kindly clarify, do you think pregnancy below age of 19 or above 40 can be labeled as low risk.

Page 4 or 5, Methods:

Study Population

line 74

Thank you for the comments.

By low-risk sample, we meant women in the low-risk conditions associated with maternal hemorrhage.

We edited this section to address your comments; the revised paragraph is in bold font.

Table -1: Age as variable should also have a reference for comparison

Page 9, Table 1

Both ways are correct, but typically, in a large dataset, this is how age is analyzed and defined in multiple logistic regressions to find if a one year increase in age either increases or decreases the risk of an event which in this case was PPH.

Recommendations and conclusions have to be well defined. References quoted in these sections are not appropriate.

Page 13-14

We added more information to these two sections.

The recommendations provided mostly were based on evidence-based practice, which is why we cited this section.

 Also, the journal asks for almost 40 references to be accepted for publication.

Reviewer 3 Report

I have reviewed a paper entitled A Nationwide Study of the “July effect” Concerning Postpartum Hemorrhage and its Risk Factors at Teaching Hospitals 3 across the United States. It looks fascinating and exciting.

The manuscript's methodology, particularly the data analysis, could be improved. To improve the manuscript, a few major issues must be addressed by all authors.

The impact of the July effect should be highlighted first in the introduction. The terminology of the July effect needs to be expanded, and it would be ideal if the authors could relate the situation to the existing theory of care.

There is a typo in a sentence line 41-42, line 54-55, and line 68-69 that needs to be corrected.

The current study is based on a previous study that used comparable data. In what ways did the previous study's findings aid in the development of the current study?

What was the study's goal: to find a correlation or an association between PPH and the "July effect"?

Could you elaborate on the data collection process based on the study design? How many times were the patients followed up? Is it correct that the study was a retrospective cohort study?

The study involved big data analysis, but readers are still curious about how the sample was selected in the first place and how the sampling method was used in the study.

On what basis was the maternal age classified?

It is preferable if the authors can specify the dependent and independent variables of each multivariate analysis. Each discovery should include the crude OR value. Each tested model must be confirmed using model fitness analysis.

According to the study's title, the authors attempt to prove the hypothesis of the association between the July effect and PPH and its risk factors. However, the findings tables revealed the opposite. The authors must clearly explain what the dependent and independent variables are in the analysis. If they want to look for the July effect, the time categories should be dependent. Because there are three categories (as stated in the manuscript), the analysis should use multinomial logistic regression.

We believe that a new analysis would yield different results. As a result, the discussion will be revisited once new findings are presented.

The study's title can be shortened to "July effect" in Postpartum Haemorrhage and its Risk Factors at Teaching Hospitals Across the United States.

Author Response

Reviewer 3

Reviewer Comment/Suggestion

Page and line number of comment to be addressed

Explanation of how it was addressed by  the authors

The impact of the July effect should be highlighted first in the introduction. The terminology of the July effect needs to be expanded, and it would be ideal if the authors could relate the situation to the existing theory of care.

Page 4, introduction, last paragraph

Thanks for your comments.

We edited the last paragraph of the introduction to address your concern.

Using the theory of care is a good idea. Although we analyze the big data, we still do not have enough evidence for the exact cause of PPH to support our points. To use the existing theory of care, we need to perform primary research to provide more evidence and better support our assumptions.

There is a typo in a sentence line 41-42, line 54-55, and line 68-69 that needs to be corrected.

Page 4-5, introduction, and methods

We tried to guess and correct those typos.

The current study is based on a previous study that used comparable data. In what ways did the previous study's findings aid in the development of the current study?

Page 4, introduction,  last paragraph

Thanks for pointing that out.

We edited this section to clarify the objective of the study.

Reviewer 4 Report

This is a well written study to understand the correlation between ‘July effect’ and the risk of postpartum hemorrhage (PPH) . It is interesting to note that the hypothesis is based on the months when inexperienced trainees are recruited.

Minor Comments:

1.     The conclusions are derived from a 2018 dataset with a large number of patients. Inorder to circumvent longitudinal samples, I wonder if approaches such as bootstrap sampling would provide stronger conclusions to differentiate risk between July effect and non-july effect patients

2.     Page 9: appendix 1: The column width of the “description” column may require adjustment to make it readable

3.     White space between line 305 and 306 may not be necessary

4.     Line 244: conclusion may have upper case C to keep headings consistent

Author Response

Reviewer 4

Reviewer Comment/Suggestion

Page and line number of comment to be addressed

Explanation of how it was addressed by the authors

The study involved big data analysis, but readers are still curious about how the sample was selected in the first place and how the sampling method was used in the study.

On what basis was the maternal age classified?

Page 4-6, Methods:

Study Population, Measures/variables, statistical analysis.

Thanks for your comments.

We provided more information on the method and its subsections to address your concerns.

The added information is in bold format.

The maternal age was classified based on age categorization in previous studies.

It is preferable if the authors can specify the dependent and independent variables of each multivariate analysis. Each discovery should include the crude OR value. Each tested model must be confirmed using model fitness analysis.

Page 4-6, Methods:

Study Population, Measures/variables, statistical analysis.

We specified dependents and independent variables as suggested.

All the information regarding variables, measures and data analysis is provided under methods and its subsections.

According to the study's title, the authors attempt to prove the hypothesis of the association between the July effect and PPH and its risk factors. However, the findings tables revealed the opposite. The authors must clearly explain what the dependent and independent variables are in the analysis. If they want to look for the July effect, the time categories should be dependent. Because there are three categories (as stated in the manuscript), the analysis should use multinomial logistic regression.

Page 9, Table 1

We provided information regarding dependent and independent variables in the methods section.

Thanks again for your suggestions.

Round 2

Reviewer 3 Report

Thank you for the manuscript update. It appears to be improved, and the correction is adequate and satisfactory. Congratulations.